# A Simple Deposition Model for Debris Flow Simulation Considering the Erosion–Entrainment–Deposition Process

Seungjun Lee [1], Hyunuk An [1,*], Minseok Kim [2], Hyuntaek Lim [3] and Yongseong Kim [3]

1   Department of Agricultural and Rural Engineering, Chungnam National University, Daejeon 34134, Korea; sjlee94@o.cnu.ac.kr
2   Geologic Environment Division, Korea Institute of Geoscience and Mineral Resources, Daejeon 34132, Korea; minseok_kim@kigam.re.kr
3   Department of Regional Infrastructure Engineering, Kangwon National University, Chuncheon 24341, Korea; voku93@hanmail.net (H.L.); yskim2@kangwon.ac.kr (Y.K.)
*   Correspondence: hyunuk@cnu.ac.kr; Tel.: +82-42-821-5797

**Abstract:** This study aimed to determine the depositional effect and improve the identification of debris flow risk zones. To accomplish this goal, we developed a two-dimensional debris flow model (Deb2D) based on a hyperbolic conservation form of the mass and the momentum balance equation with consideration of the erosion–entrainment effect as well as the depositional effect. In this model, we implemented a widely-used rheological equation—the Voellmy equation—and a quadtree adaptive grid-based shallow-water equation. This model was applied to two study sites to assess the depositional effect. The impact area, volume of soil loss, maximum velocity, inundated depth, and erosion depth resulting from the debris-flow modeling were compared with the field data. The simulation results with/without deposition were evaluated using the receiver operating characteristic method. The implementation results of the erosion–entrainment model with deposition showed superior accuracy when estimating the damage range and flow time.

**Keywords:** landslides; debris flow; depositional effect; erosion–entrainment

## 1. Introduction

Debris flows, defined as gravity-driven sediment mixtures with surface and subsurface flow in mountainous areas, are one of the most serious natural disasters. Additionally, it can be difficult to predict the extent and scale of their resultant damage. In particular, debris flows are capable of rapidly transporting large volumes of sediment and large boulders over long distances, making them destructive and dangerous [1,2] according to local properties such as the geology, topography, and saturated soil by rainfall. Therefore, several studies have attempted to analyze and predict the flow and accumulation process of debris flows using various methods including experimentation [3,4], monitoring [5–9], and numerical modeling [10–15]. In particular, many notable numerical studies of debris flow have been performed in recent decades [16,17] and they have provided a better and more detailed understanding of debris flow mechanisms, and their results reflect the suggested theory.

Numerical models are useful for analyzing the flow of difficult-to-implement mixtures such as debris flows. Various phenomena occur during the debris flow process, of which erosion, entrainment, and deposition are typical. Here, the erosion process refers to the removal of the topsoil, and the entrainment refers to the absorption of the topsoil removed through the erosion process. Therefore, erosion–entrainment is an essential factor when predicting and estimating the magnitude of damage because this effect increases the initial volume of the debris flow by 10–50 times [18,19]. In addition, deposition refers to a phenomenon in which flowing sediment movement stops and accumulation begins after reaching a specific condition; deposition is closely linked to the travel distance [2,20]. Therefore, deposition is a significant factor in predicting and calculating the extent of

damage caused by debris flows. Therefore, erosion, entrainment, and deposition must all be considered in numerical modeling to accurately analyze and predict the extent and scale of damage caused by debris flows.

Several one-dimensional (1D) and two-dimensional (2D) models have been proposed, and 2D simulations have provided clear insights into the dynamic behavior of debris flows [21–25]. In addition, several erosion–entrainment models have been developed based on different theories and research on the erosion and entrainment phenomena that occur during the debris flow process has been actively conducted in recent decades [10,19,26–29].

In several studies [30–33], the erosion–deposition process was implemented using the sediment concentration of debris flows and physical parameters. Pudasaini and Fischer [34] developed the erosion and deposition algorithm through a full physical mechanism based on a two-phase system. However, setting these physical parameters (e.g., grain size, particle diameter, internal friction angle) requires many field surveys. In addition, these characteristics are different for each basal, and it is limited to being defined as a representative value for one basin. Therefore, the applicability of models using physical parameters is still challenging. Medina et al. [10] developed a "stop-and-go" mechanism to estimate damaged areas. However, this mechanism is limited in its ability to implement the deposition process because it simply determines at what point a debris flow will stop, and in addition, its applicability is limited because a detailed explanation of the involved mechanisms was not included. Since the deposition process is relevant in predicting the extent of damage caused by debris flow, several studies on the deposition process have been conducted. However, since most algorithms use physical parameters based on physical mechanics, their usefulness is low for various basal and rainfall situations. Therefore, to predict/respond instantaneously to debris flow that occurs in landslide-prone areas, additional research on the deposition process algorithm based on the conceptual system is needed.

This study developed a simple deposition model for debris flow simulation with a combined erosion–entrainment model. This model refers to the algorithms developed in Medina et al. [10] and Frank et al. [29] that assume that the erosion–entrainment process occurs when the basal shear stress exceeds a specific value, and these algorithms well-simulate the erosion–entrainment process at real-scale debris flow events. Therefore, the developed model simulates the erosion, entrainment, and deposition process based on the algorithms of Medina et al. [10] and Frank et al. [29]. To verify the necessity and importance of the deposition effect, the past two debris flow events (the 2011 Mt. Umyeon debris flow and the 2020 Gokseong-gun debris flow) were selected and analyzed.

Since the field survey data are abundant, the 2011 Mt. Umyeon landslides, which are suitable for the analysis of debris flow, were selected as the study events and the parameters of the deposition algorithm were analyzed. In addition, a recent debris flow event, which occurred in 2020 in Gokseong-gun and killed six people, was simulated to verify the versatility and accuracy of the algorithm developed in this paper. To evaluate the analytical results, the simple method and the receiver operating characteristic (ROC) methods were used to quantitatively analyze the area damaged by the debris flow.

## 2. Methods

### 2.1. Governing Equation and Numerical Model

This study used the Deb2D model developed by An et al. [25] to test the deposition model's performance. The Deb2D model's governing equation is a hyperbolic conservation form of the mass and the momentum balance equation is expressed as follows:

$$\frac{\partial \mathbf{q}}{\partial t} + \frac{\partial \mathbf{f}}{\partial x} + \frac{\partial \mathbf{g}}{\partial y} = \mathbf{s}, \tag{1}$$

where $t$ denotes time; $x$ and $y$ are Cartesian coordinates; and **q**, **f**, **g**, and **s** are vectors representing conserved variables, fluxes in the $x$ and $y$ directions, and source terms, respectively. The vectors can be written as:

$$\mathbf{q} = \begin{pmatrix} h \\ hu \\ hv \end{pmatrix}, \mathbf{f} = \begin{pmatrix} hu \\ hu^2 + gh^2/2 \\ huv \end{pmatrix}, \mathbf{g} = \begin{pmatrix} hv \\ huv \\ hv^2 + gh^2/2 \end{pmatrix}, \mathbf{s} = \begin{pmatrix} ED \\ S_{gx} - S_{fx} \\ S_{gy} - S_{fy} \end{pmatrix}, \quad (2)$$

where $h$ is the depth of the debris-flow mixture; $u$ and $v$ are the depth-averaged velocity components in the $x$ and $y$ directions, respectively; $g$ is the acceleration due to gravity; $ED$ is the entrainment ($ED > 0$) and deposition rate ($ED < 0$); $S_{gx}$ and $S_{gy}$ represent the gravitational acceleration in the $x$ and $y$ directions, respectively; and $S_{fx}$ and $S_{fy}$ represent the driving friction in the $x$ and $y$ directions, respectively.

The rheological equation is one of the most significant factors and it dominates the behavior of debris flows. Several rheological equations have been used to simulate debris flows and Deb2D can consider the Voellmy, Bingham, and Coulomb-viscous rheological equations. This study used the Voellmy equation because it demonstrated good performance in several previous studies (e.g., [10,35]) while having fewer parameters than other rheological equations. The Voellmy rheology equation is expressed as follows:

$$S_{fx} = \left( \mu g h + \frac{g|u|^2}{\xi} \right), S_{fy} = \left( \mu g h + \frac{g|v|^2}{\xi} \right), \quad (3)$$

where $\mu$, which represents the Coulomb friction coefficient, dominates the deceleration behavior when the flow is slow and $\xi$, which represents the turbulent friction coefficient, prevails when the flow is rapid [29,36].

Numerical models based on shallow-water-type governing equations frequently suffer from the imbalance that occurs between the slope of the water depth and the slope on irregular terrain. This imbalance may cause unphysical perturbations and fluctuations in the simulation near shocks or wet–dry transitions. Deb2D implements the hydrostatic reconstruction technique proposed by Audusse et al. [37], which has successfully been applied to the quadtree adaptive grid-based shallow-water equation. For a detailed numerical discretization and scheme, refer to An et al. [25,38]. In addition, Figure 1 depicts a flowchart of the Deb2D model.

### 2.2. Deposition Model with Erosion–Entrainment Phenomena

Many previous studies have identified the importance of the erosion–entrainment phenomena that occur as debris flows progress [10,25,29,39,40]. The treatment of the erosion–entrainment process in numerical models can be classified into two methods. The first approach assumes that the erosion–entrainment occurs at a constant rate [10,29] and the second approach considers that the erosion-entrainment process's velocity is determined by the state of the debris flow [10,27,28]. This study used the first approach, which demonstrates the erosion–entrainment process based on physics using basal shear stress. On the other hand, the second approach simulated the erosion–entrainment process using full conceptual/experimental equations that have been limited to demonstrate the erosion–entrainment action reasonably. According to Medina et al.'s [10] static and Frank et al.'s [29] algorithms, the erosion and entrainment process occurred at a constant velocity when the basal shear stress was higher than a critical condition during the flow. This study assumed that the deposition occurred at a constant velocity when the shear stress was lower than a specific condition. The erosion–entrainment–deposition mechanism $ED$ is expressed as follows:

$$ED(x,y,t) = \begin{cases} \frac{dz}{dt}_{eros.} & \text{if } \tau > \tau_{eros.} \\ -\frac{dz}{dt}_{depos.} & \text{else if } \tau < \tau_{depos.} \end{cases}, \quad (4)$$

$$h_{\max}(x, y, 0) = \begin{cases} \frac{dz}{d\tau}(\tau - \tau_{eros.}) = \frac{dz}{d\tau}(\rho g h s - \tau_{eros.}) & \text{if } \tau > \tau_{eros.} \\ 0 & \text{else} \end{cases}, \qquad (5)$$

where $dz/dt_{eros.}$ is a constant erosion–entrainment rate; $dz/dt_{depos.}$ is a constant deposition rate; $\tau$ is the basal shear stress; $\tau_{eros.}$ and $\tau_{depos.}$ are the critical shear stress of erosion and deposition, respectively; $h_{\max}(x, y, 0)$ is the initial height of the erosion–entrainment layer at position $(x, y)$. $dz/d\tau$ is the average potential erosion depth; $\rho$ is the mass density; and $s$ is the channel slope. Figure 2 briefly illustrates this algorithm. In addition, this algorithm's performance, which well simulated the erosion–entrainment–deposition process, has been verified based on laboratory experiments conducted by Lim et al. [41].

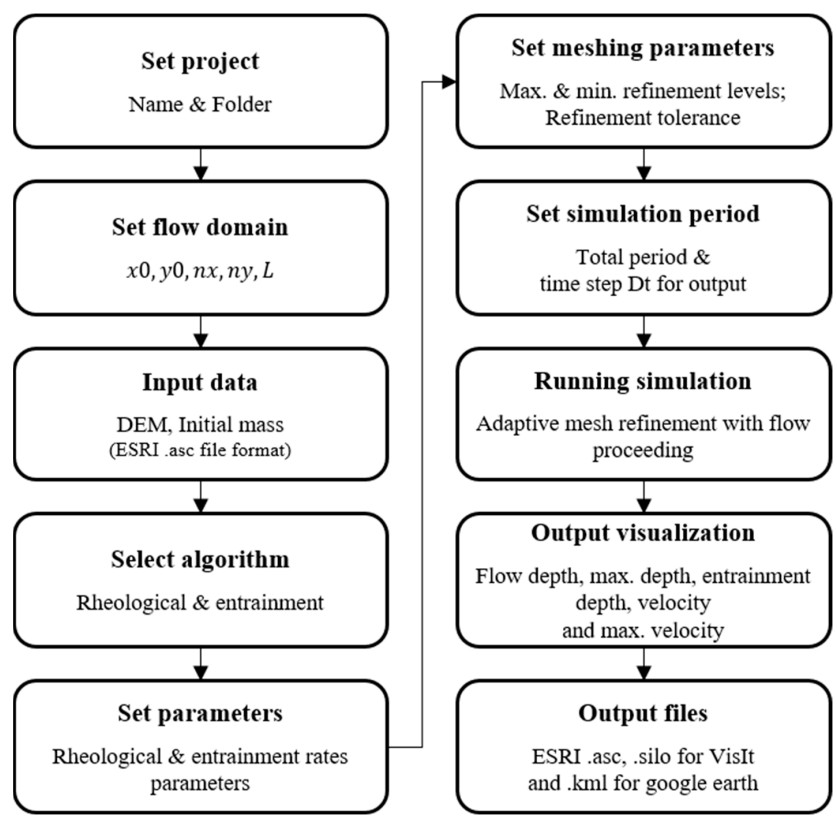

**Figure 1.** Flowchart of the Deb2D model (modified from An et al. [25]).

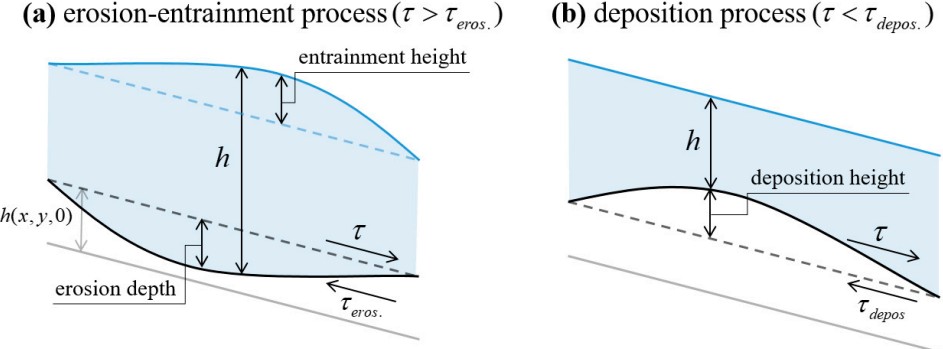

**Figure 2.** Processes of (**a**) erosion–entrainment and (**b**) deposition.

The Voellmy rheological equation, which simulates the sediment flow process, uses the parameters calibrated in Lee et al. [40], who analyzed the parameters of the Voellmy equation on the 2011 debris flow in Mt. Umyeon. To determine the importance of deposition, only the deposition algorithm's parameter was changed to compare the simulation results.

*2.3. Analysis Method*

Godt et al. [42] and Cepeda et al. [43] used the ROC method to quantitatively analyze the debris flow's process and damage. Therefore, we used the ROC method to analyze qualitative data including the range of the impact areas from the simulations. In particular, the ROC method's accuracy factor was used. The accuracy analysis of the ROC is as follows:

$$\text{Acc.}_R = \frac{\text{D.A} + \text{nD.A}}{\text{C.A}} \tag{6}$$

where Acc., which is the accuracy, is a value in the range 0–1 and 1 denotes 100% accuracy and 0 denotes 0% accuracy. D.A is the area that successfully implemented the actual damaged area; nD.A denotes the area that was not damaged and also indicates that the simulation results showed that there was no damage; and C.A is the calculation area (Figure 3). For details regarding the ROC method, please refer to Godt et al. [42] and Cepeda et al. [43].

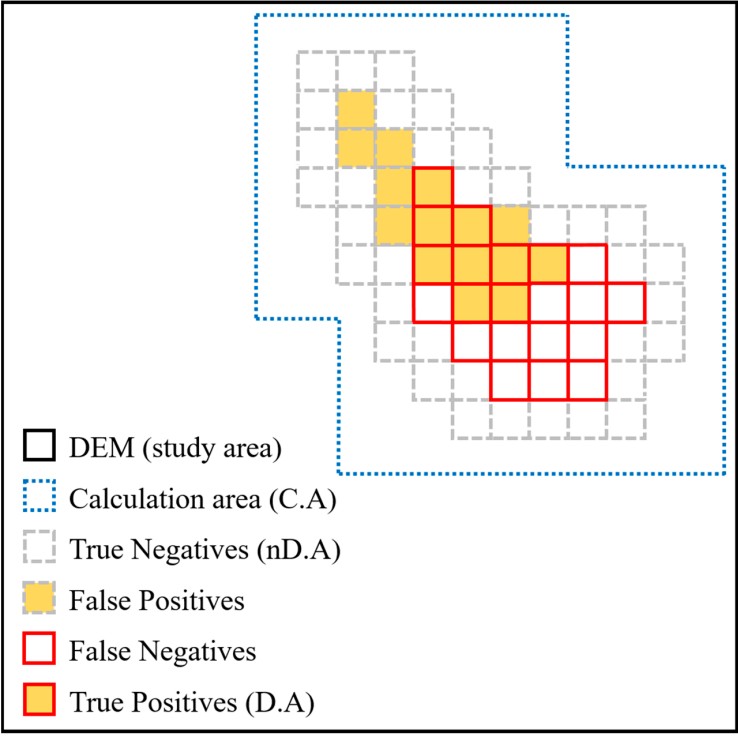

**Figure 3.** Definition of variables for the estimation of discrete classifiers for spatially distributed values by the ROC method (modified from Godt et al. [42] and Cepeda et al. [43]).

The accuracy of the quantitative results, which include the sediment height, flow velocity, and relocated sediment during the debris flow process, were calculated as follows [40]:

$$\text{Acc.}_L = \begin{cases} 0 & \text{if Sim.val.} \geq 2 \times \text{Obs.val.} \\ 1 - \frac{|\text{Obs.val.} - \text{Sim.val.}|}{\text{Obs.val.}} & \text{else} \end{cases}, \tag{7}$$

where Obs.val. and Sim.val. are the observed and simulated values, which include the inundated depth, flow velocity, and volume of eroded sediment. The overall accuracy TotalAcc. was calculated as follows:

$$\text{TotalAcc.} = \frac{\sum\limits_{i=1}^{N} (\text{Acc.}_i \times w_i)}{\sum\limits_{i=1}^{N} w_i}. \tag{8}$$

where Acc.$_i$ is the accuracy calculated using Equations (6) and (7); $N$ is the number of all comparisons; and $w_i$ is the weight of each comparison. Here, the weight of each comparison was determined by referring to the reliability of the field data and the coefficient proposed by Cepeda et al. [43]. Therefore, using Equation (8), we attempted to comprehensively evaluate the flow process and impact areas that appeared after the flow.

## 3. Study Event

Two debris flow events that occurred in the Republic of Korea were selected to validate the deposition model proposed in this study. The first event was the debris flow that occurred in Mt. Umyeon in 2011, as the field data have been investigated extensively by many previous researchers [25,40,44–47] due to its severe damage and location at the center of the capital city (Figure 4). In particular, field survey data are essential for parameter correction and back-analysis and are also useful when validating a numerical model. The second chosen event was the debris flow that occurred in Gokseong-gun in August 2020. The extent of the damage after the debris flow has been investigated directly and analyzed, so this event was used to test the model performance.

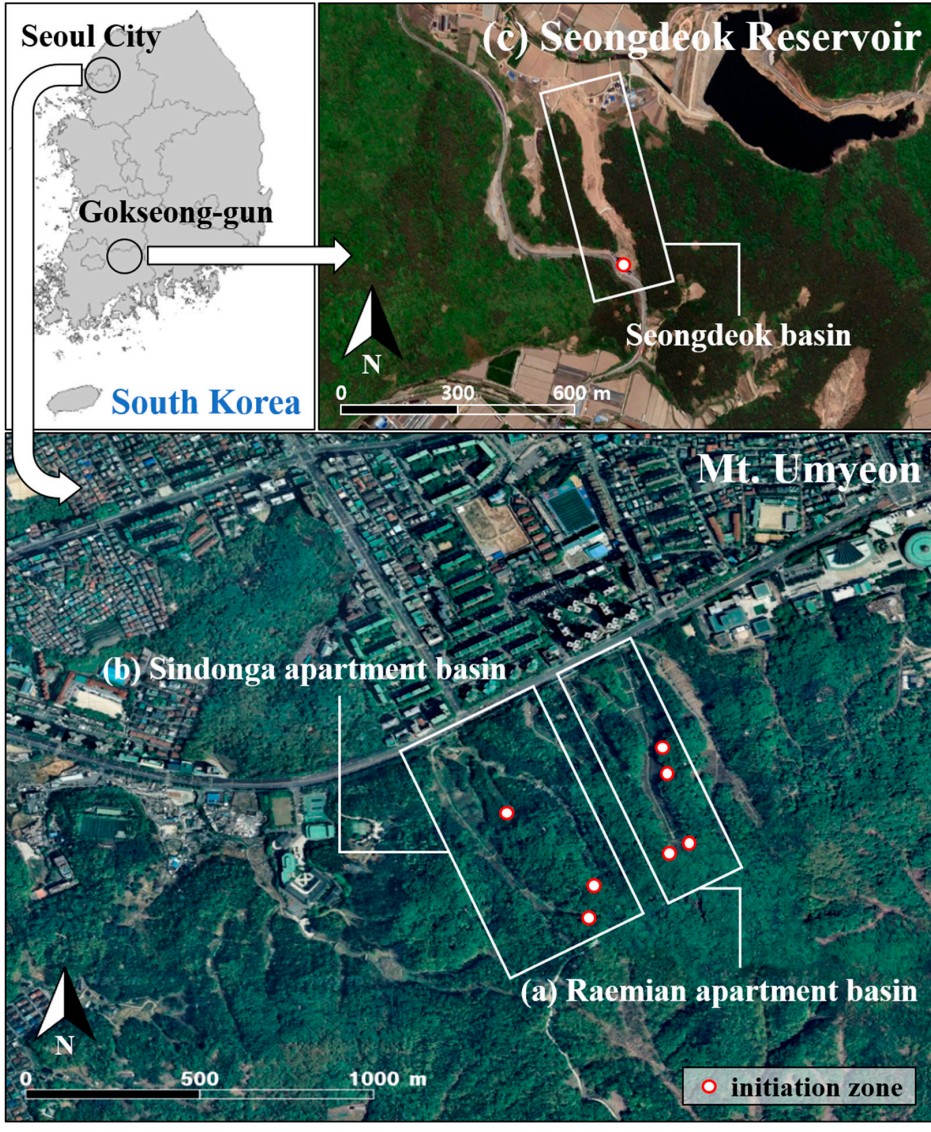

**Figure 4.** The study event location: the debris flows at Mt. Umyeon in 2011, (**a**) Raemian and (**b**) Sindonga apartment basins, and (**c**) Gokseong-gun, Seongdeok basin in 2020.

### 3.1. 2011 Mt. Umyeon Debris Flows

In the Umyeon Mountain area located in Seoul City, 16 shallow landslides and two runout debris flows occurred due to intense rainfall (rainfall amount: 500 mm/day; maximum rainfall intensity: 80 mm/h) on 26 and 27 July 2011 [48]. The bedrock in this area is primarily composed of Precambrian banded biotite gneiss and granitic gneiss and the study area had an average 34° slope [48]. The soil depth within the hillslopes in the study area was in the range of 1–4 m (average 2 m) [48]. When the debris flow occurred due to torrential rainfall, there was significant damage to people and property in the residential area near Mt. Umyeon and the sediment deposited on the nearby road paralyzed traffic.

Debris flows translated by shallow landslides occurred at four points near the top of the mountain within the Raemian catchment (Figure 4a) and at three points in the Sindonga catchment (Figure 4b). The observed channel lengths in the Raemian and Sindonga basins were reportedly 606 m and 664 m, respectively. The volume of soil loss was evaluated by comparing the digital elevation models using light detection and ranging (LiDAR DEMs) before and after the debris flow events. The soil loss volumes were 25,940 m$^3$ and 21,070 m$^3$ in the Raemian and Sindonga basins, respectively. The maximum velocity of the debris flows, which were analyzed using CCTV and a car black box, was estimated to be approximately 28 m/s and 18 m/s at the Raemian and Sindonga apartment blocks, respectively. It was confirmed that the Raemian and Sindonga apartments were directly damaged to the 3rd floor (approximately 10 m) and the 2nd floor (8 m), respectively, due to the debris flows.

In this study, a LiDAR DEM (1 m × 1 m) was used to input data for the topography. The scale of the collapse spots was determined based on the difference in the LiDAR DEMs before and after the landslide. Therefore, the initially measured landslide volumes were 350 m$^3$ for both catchments.

Using Equations (6)–(8), the results were compared and analyzed using field data. The criteria were (1) the impact area; (2) the volume of soil loss; (3) the inundated depth observed near the apartment blocks; and (4) the debris flows' maximum velocity.

### 3.2. 2020 Gokseong-Gun Debris Flows

In the Seongdeok reservoir, Gokseong-gun, which is located in South Jeolla Province, debris flows recently occurred due to intense rainfall (rainfall amount: 277 mm/3 days; maximum rainfall intensity: 52 mm/h) 5–7 August 2020 (Figure 4c). The length of the Seongdeok basin is 678 m, which is similar to the two Mt. Umyeon basins. The debris flow caused by the torrential rainfall was rapid, whereas the flow velocity of the debris flow was slower than that of the 2011 Mt. Umyeon debris flow. Nevertheless, the volume of the slope collapse that occurred near the road was quite large and it caused considerable damage to rural areas and rice paddies in the lower part of the watershed, and five fatalities occurred [49].

According to the on-site survey, a large-scale collapse occurred on the slope facing the road and the collapsed area had an average slope of 35°. According to Choi et al. [49] and the field survey, the erosion depth was <2.5 m and the volume of soil loss was evaluated at ~30,000 m$^3$. There is a river in the lower region, therefore, some debris flows into the river. However, since this study simulated the debris flow based on a one-phase-/-layer, the implementation for such inflow into the river was limited. The principal flat bottom valleys in Gokseong-gun are composed of deeply weathered soils [50]. The characteristic weathering profile of these granite and binary rocks are gravel and laminated gravel tens of meters thick with irregularly oriented crack areas under the loose weathering material [49]. A DEM (5 m × 5 m) was used as the input data for the topography. The data on the collapsed area were constructed by referring to the digital surface model (DSM) built by drones after the debris flow event and the volume was set to approximately 14,500 m$^2$. Since this basin lacks field data, precise comparative analysis is limited so we attempted to confirm the implementation rate of the impact area using Equation (6).

## 4. Results

### 4.1. Simulation Results: Mt. Umyeon Area

Table 1 summarizes the parameters used in the catchment of the Raemian and Sindonga apartments. The Voellmy rheological equation and the erosion–entrainment–deposition model's parameters were calibrated for each basin. In addition, to reveal the importance of deposition, the parameter used in the deposition algorithm was the only one we changed ($dz/dt_{depos.}$ = 0 or 0.01).

**Table 1.** Calibrated parameters for the Deb2D debris-flow model (Voellmy rheological equation with erosion–entrainment algorithm considering deposition) at the Mt. Umyeon and Gokseung-gun debris flow events.

| Study Event | Voellmy Rheology | | Erosion–Entrainment | | | | Deposition |
|---|---|---|---|---|---|---|---|
| | $\mu$ | $\zeta$ (m/s²) | $dz/dt_{eros.}$ (m/s) | $\tau_{eros.}$ (kPa) | $dz/d\tau$ (m/kPa) | $\rho$ (kg/m³) | $dz/dt_{depos.}$ (m/s) |
| Raemian apartment basin | 0.04 | 3000 | 0.08 | 1.0 | 0.3 | 1900 | 0.01 |
| Sindonga apartment basin | 0.04 | 3000 | 0.07 | 1.5 | 0.3 | 1900 | 0.01 |
| Seongdeok basin | 0.05 | 2000 | 0.07 | 1.0 | 0.2 | 1900 | 0.01 |

Figure 5 presents the simulated maximum flow height of the debris flow according to the algorithm with/without the deposition process in both catchments. As this figure shows, there is a clear difference in the extent of the damage depending on the deposition. When the deposition process was considered, the impact area observed through the field surveys was adequately demonstrated in both catchments (Figure 5a,c). However, when the deposition process was not considered, the simulated damage caused by the debris flow occurred over a wider area because the impact area was overestimated; it appeared that it went outside of the calculation range set in the numerical model (Figure 5b,d).

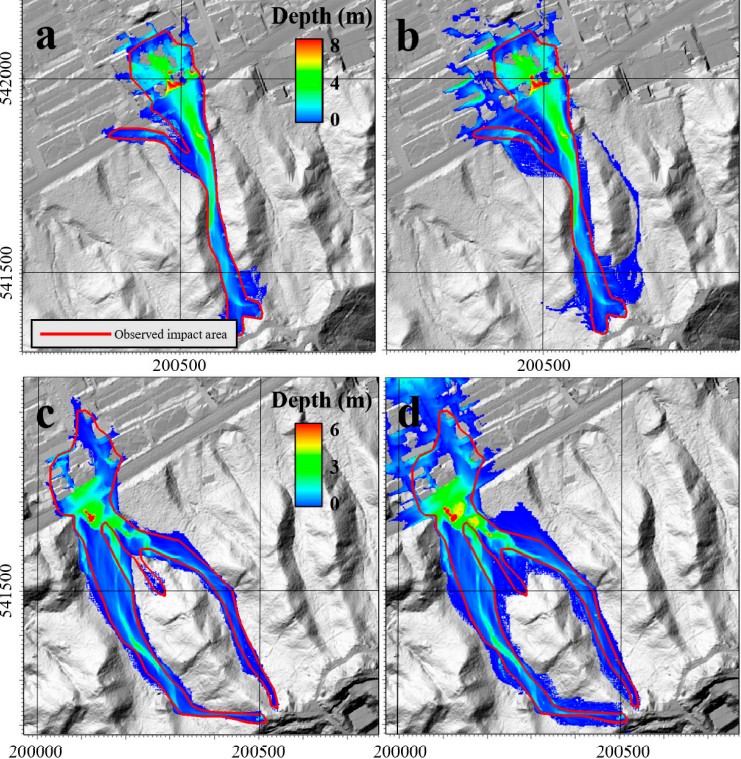

**Figure 5.** The maximum depth of the Mt. Umyeon debris flows after 5 min on the Raemian apartment basin simulation (**a**) with deposition and (**b**) without deposition; and the Sindonga apartment basin simulation (**c**) with deposition and (**d**) without deposition.

To precisely analyze this phenomenon, Figures 6 and 7 show the height of the debris flow over time in the Raemian apartment basin. Figure 6 shows the results when the deposition was considered; when the flow time reached 2 min, most of the debris flow had stopped and was deposited. The damage range did not spread any farther. However, according to the simulation results in Figure 7, which did not consider the deposition process, the flow of sediment did not stop over time and the impact area was unphysically expanded. In addition, we attempted to determine the effect of the parameter change in the deposition algorithm on the estimation of the damaged area (Figure 8). A larger $dz/dt_{depos.}$ meant that the deposition process was larger and the simulated impact area was very small. Conversely, when the parameter was closer to 0, the impact area was wider.

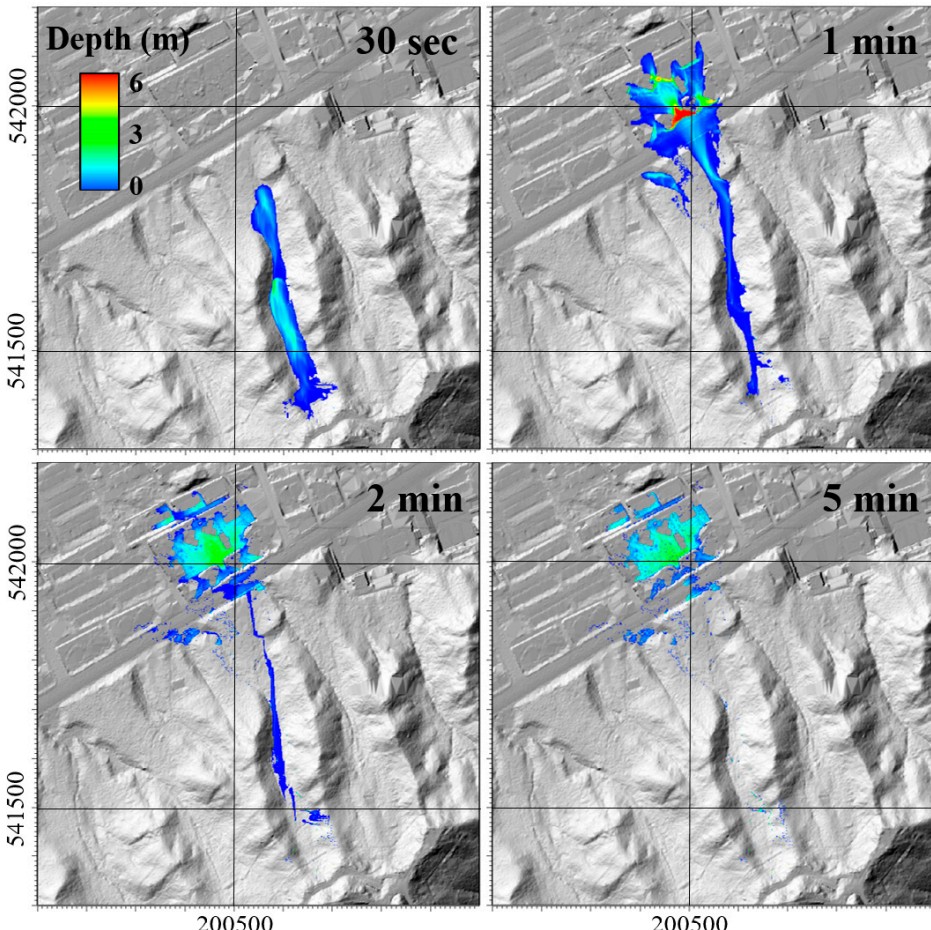

**Figure 6.** Simulated flow depth at the Raemian apartment basin over time with deposition where $dz/dt_{depos.} = 0.01$.

Figure 9 shows the simulation results of the erosion depth for the two catchments. Figure 9a,d depict the observations for each catchment, which were constructed using LiDAR DEMs before and after the occurrence of the debris flow. According to the results, in both basins, there was no critical difference with or without the deposition process. However, when the deposition was accounted for, as shown in Figure 9b,e, the phenomenon of widespread erosion that was demonstrated in Figure 9c,f was reduced, which is contrast from the results from the field data. In addition, it showed that there was a considerable difference in the volume of soil loss according to the with/without deposition process. Therefore, this result revealed that the erosion depth, which was equivalent to the topography change, was simulated as relatively similar to the results shown in Figure 9b,e in which the deposition process was considered.

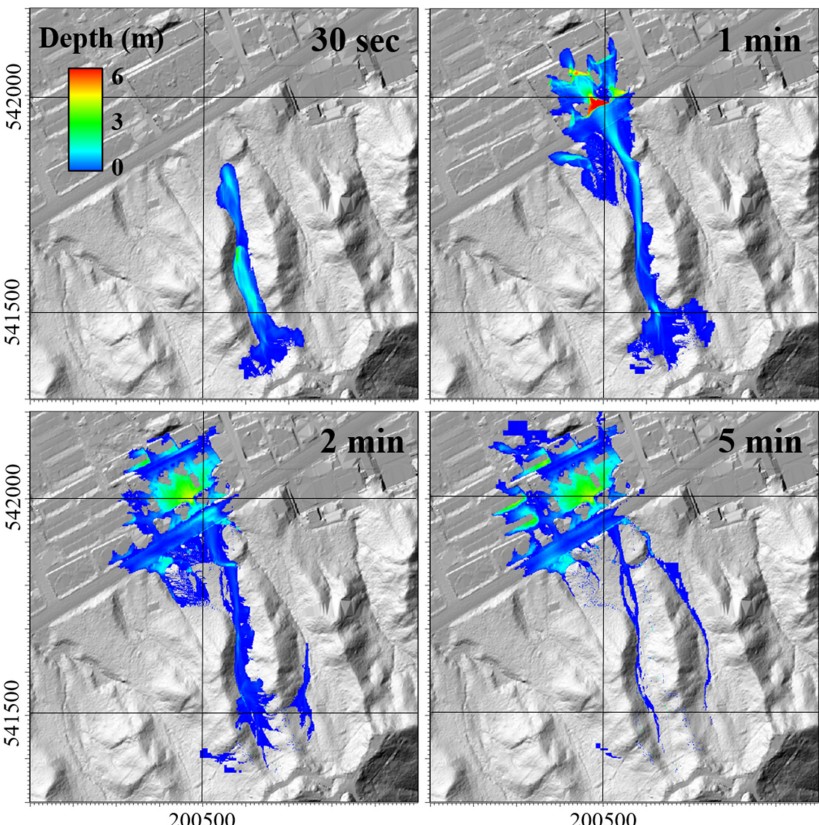

**Figure 7.** Simulated flow depth at the Raemian apartment basin over time without deposition where $dz/dt_{depos.} = 0$.

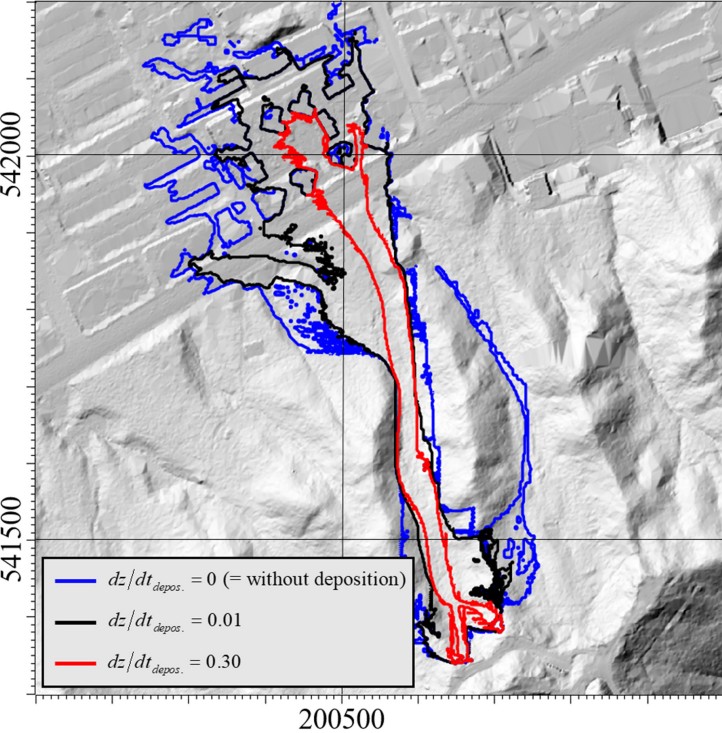

**Figure 8.** Spreadability of debris flow according to the change in parameter $dz/dt_{depos.}$ at the Raemian apartment basin.

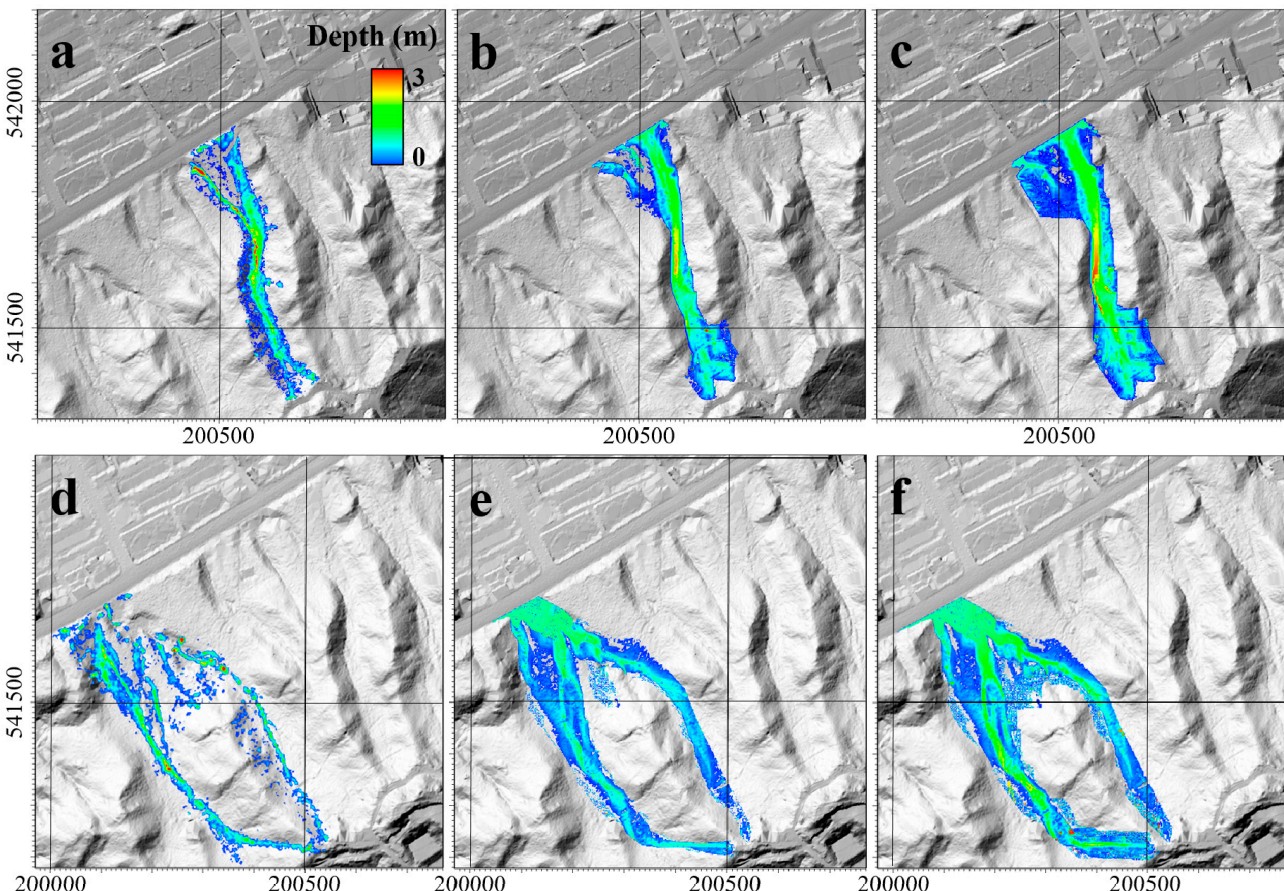

**Figure 9.** Erosion depth of the Mt. Umyeon debris flows after 5 min in the Raemian apartment basin (**a**) observed, (**b**) with deposition, where $dz/dt_{depos.}$ = 0.01 and (**c**) without deposition, where $dz/dt_{depos.}$ = 0; and the Sindonga apartment basin (**d**) observed, (**e**) with deposition, where $dz/dt_{depos.}$ = 0.01 and (**f**) without deposition, where $dz/dt_{depos.}$ = 0.

Table 2 summarizes the simulation results for the debris flow in the two catchments. In the Raemian apartment basin, the volume of soil loss analyzed with/without the deposition process was 25,260 m$^3$ and 47,390 m$^3$, respectively. When deposition was considered, the observed value of 25,940 m$^3$ was well-simulated. The amount of soil loss analyzed with/without deposition in the Sindonga apartment basin was 21,900 m$^3$ and 48,170 m$^3$, respectively. In addition, when the deposition process was accounted for, the observation of 21,070 m$^3$ was well-simulated. As such, the analysis results for the volume of soil loss revealed large differences depending on the deposition process. However, the analysis results for the maximum damage height of the debris flow observed near the Raemian and Sindonga apartments and the maximum flow velocity of the debris flow observed near the roads were similar, regardless of the deposition process. In the case of the Raemian apartment basin, the inundated depths analyzed with/without deposition were 10 m and 11 m, respectively, and they matched the observed values of 9–11 m. In the Sindonga apartment basin, the inundated depth was implemented to be 6 m, regardless of the deposition process and this value was consistent with the observation of 6–8 m. In addition, the maximum flow velocity was analyzed to be 16 m/s, regardless of the deposition process, which was consistent with the observed value of 18 m/s. However, in the Raemian apartment basin, the maximum flow velocity was analyzed to be 19 m/s regardless of the deposition process, which was slow compared to the observation of 28 m/s.

**Table 2.** 2011 debris flow simulation results for the Raemian and Sindonga apartment basins with/without deposition process.

| Study Event | With/Without Deposition | Volume of Soil Loss (m³) | Inundated Depth (m) | Maximum Velocity (m/s) |
|---|---|---|---|---|
| Raemian apartment basin | with- | 25,260 | 10 | 19 |
| | without- | 47,390 | 11 | 19 |
| | observation | 25,940 | 9–11 | 28 |
| Sindonga apartment basin | with- | 21,900 | 6 | 16 |
| | without- | 48,170 | 6 | 16 |
| | observation | 21,070 | 6–8 | 18 |

Table 3 summarizes the implementation rate results of the flow process and impacted area analyzed using Equations (6)–(8). As a result, when the deposition was considered at the Raemian apartment basin in Table 3, TotalAcc. was 0.914, and when the deposition was not considered, it was analyzed to be 0.779. At the Sindonga apartment basin, when the deposition was considered, TotalAcc. was 0.917; when the deposition was not considered, it was analyzed as 0.753. Overall, the simulation accuracy was 0.13–0.17 higher when considering the deposition process. In particular, considering the deposition, impact area and the amount of soil loss resulted in a more accurate prediction. Moreover, the simulation without deposition demonstrated the worst performance in terms of "volume of soil loss".

**Table 3.** Analysis results reported by the simple and ROC methods when analyzing the debris flows in the 2011 Raemian and Sindonga apartment basin simulations according to with/without deposition action.

| Study Event | With/Without Deposition | Acc. | | | | Total Acc. |
|---|---|---|---|---|---|---|
| | | Impact Area | Volume of Soil Loss | Inundated Depth | Maximum Velocity | |
| | $w_i$ | 3 | 1 | 2 | 1 | |
| Raemian apartment basin | with- | 0.914 | 0.974 | 1.000 | 0.679 | 0.914 |
| | without- | 0.868 | 0.173 | 1.000 | 0.679 | 0.779 |
| Sindonga apartment basin | with- | 0.857 | 0.961 | 1.000 | 0.889 | 0.917 |
| | without- | 0.794 | 0.000 | 1.000 | 0.889 | 0.753 |

*4.2. Simulation Results: Seongdeok Area*

Table 1 summarizes the parameters used to analyze the Seongdeok basin. The Voellmy rheological equation and the erosion–entrainment–deposition model's parameters were also calibrated for this basin based on field survey data. The Voellmy rheology parameters that control the fluidity of debris flow were calibrated considering that the rainfall intensity and total rainfall were less than that for the previous two basins. Similar to the Mt. Umyeon landslides, they were implemented with/without the deposition process ($dz/dt_{depos.}$ = 0 or 0.01). In the case of the Seongdeok basin, the simulated results and the observation data that could be compared and analyzed included only the damaged range. Therefore, using Equation (6), we assessed the implementation rate (accuracy) of the damage range that was analyzed using the algorithm developed in this study.

Figure 10 shows the damaged area observed by drones and on-site surveys as well as the simulation results for the maximum flow depth of the debris flow after 5 min, and the river was briefly displayed. As shown in Figure 10a, the impact area was well-depicted when the deposition was considered. However, when the deposition process was not considered, the damaged area was much wider than what was observed, as shown in

Figure 10b. In the numerical model, it was difficult to implement the effect of the river and rice fields. Therefore, regardless of the deposition process, the damage that occurred across the river was not observed in the field. This phenomenon was demonstrated precisely in Figure 11, which analyzed the debris flow height according to time. As shown in Figure 11, the river blocked some sediments. However, the system by which the sediments flowed into the river was not simulated, and it was difficult to implement the sediment flow into the rice fields that could have reduced the damage of the debris flow. However, it was sufficient to simulate and analyze the impact area of the debris flow that occurred in the Seongdeok basin.

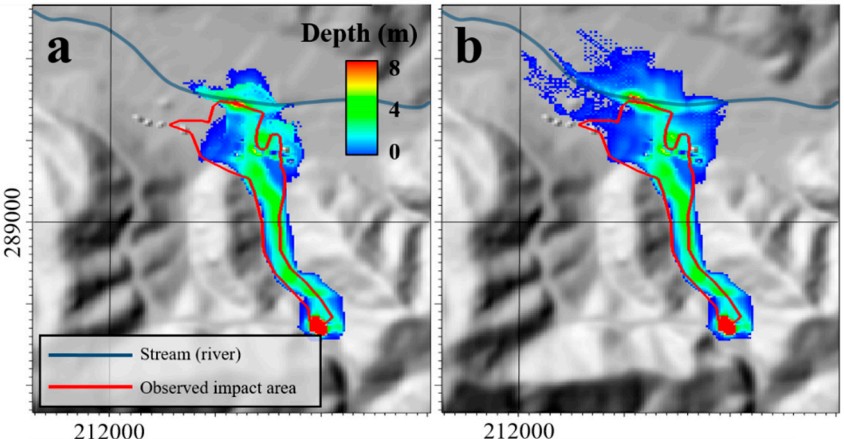

**Figure 10.** Maximum depth of the debris flow at Seongdeok basin after 5 min (**a**) with deposition and (**b**) without deposition.

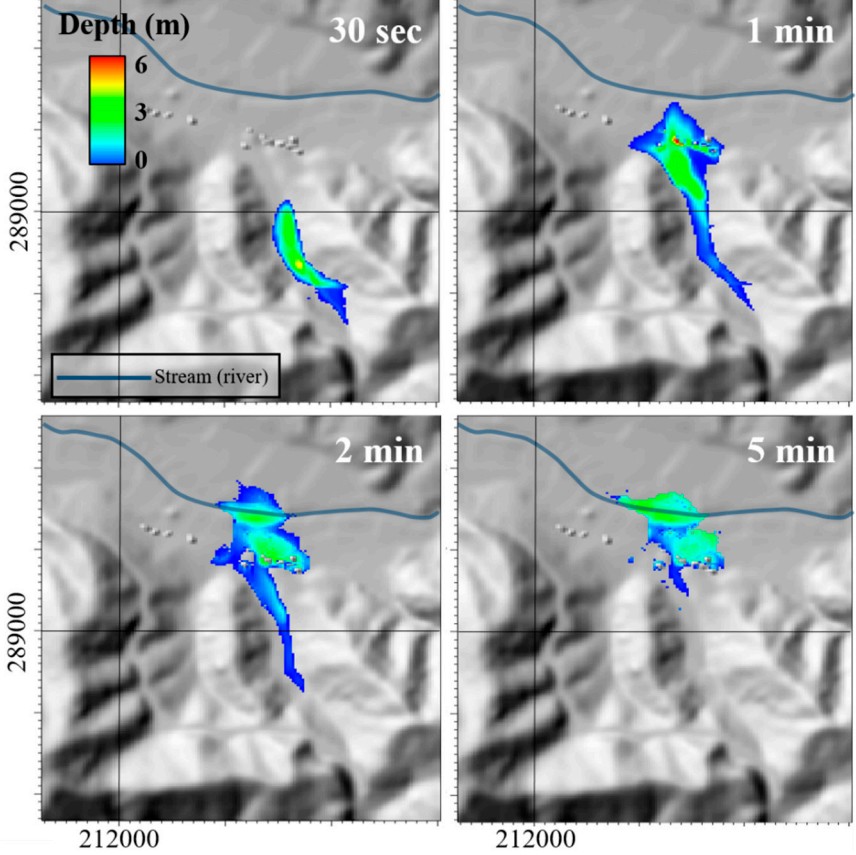

**Figure 11.** Simulated flow depth at the Seongdeok basin over time with deposition.

According to the field survey and Choi et al. [49], the amount of soil loss was analyzed to be about ~30,000 m$^3$, which is the volume calculated by considering both erosion and deposition. For the simulation result in the Seongdeok basin, when the deposition effect was considered, the analyzed soil loss was 29,600 m$^3$, which is consistent with the volume calculated in the field survey and previous studies. However, when deposition was not considered, the amount of soil loss was analyzed to be 41,980 m$^3$, which is an overestimate.

Table 4 summarizes the simulation results of the debris flow. When simulating with deposition, the impact area was analyzed with an accuracy of 0.952. The simulation without deposition demonstrated the impact area with high accuracy of 0.909. However, the volume of soil loss in the basal was overestimated considerably. In addition, the maximum flow velocity observed in the residential area (rural area) was analyzed to be 10 m/s, and in the simulation, the debris flow affected the village up to 6 m. These results revealed a lower value than that of the Mt. Umyeon debris flow event, but it was estimated to be sufficient to cause damage to the village.

**Table 4.** 2020 Seongdeok basin debris flow simulation results with deposition action.

| Study Event | With/Without Deposition | Volume of Soil Loss (m$^3$) | Inundated Depth(m) | Maximum Velocity (m/s) | Impact Area (Acc.) |
|---|---|---|---|---|---|
| | with- | 29,600 | 6 | 10 | 0.952 |
| Seongdeok basin | without- | 41,980 | 6 | 10 | 0.909 |
| | observation | ~30,000 | Unmeasured | Unmeasured | Field survey |

## 5. Discussion

Figure 5 shows the importance of the deposition process. When deposition was considered, the extent of the damage was simulated similarly to the field survey, as shown in Figure 5a,c. However, when the deposition process was not considered, as shown in Figure 5b,d, the impact area was overestimated compared to the result from the field survey. To analyze this phenomenon more precisely, the flow height of the debris flow over time was analyzed for the Raemian apartment basin, as shown in Figures 6 and 7. According to these simulation results, the deposition process stopped the flow of sediments under certain conditions, as shown in Figure 6, and caused them to be deposited; additionally, the analysis that accounted for deposition helped to estimate the flow time of the debris flow. However, when simulating the debris flow event without the deposition process, the damaged area was extended unnaturally over time, as shown in Figure 7. Therefore, to accurately estimate the area impacted by the debris flow, Figures 5 and 6 demonstrate that it is essential to consider the deposition process. However, as shown in Figure 8, it revealed that parameter calibration was necessary for the deposition algorithm, as in the Voellmy rheological equation and the erosion–entrainment model. This phenomenon, which stopped the flow of sediments, was implemented in Medina et al. [10] using the "stop-and-go" system. Since the "stop-and-go" system controls the flow and stoppage of debris flow, this system made it possible to precisely analyze the damaged area in Median et al.'s [10] study. However, a detailed explanation of the "stop-and-go" mechanism was omitted and it played a strictly different role from that in the deposition process, so its applicability was poorer than the deposition algorithm introduced in this paper.

Figure 9 shows the simulation results of the erosion depths in both catchments. Visually, there was no critical difference according to the presence/absence of the deposition process. However, when analyzed with the "volume of soil loss" in Table 2, it was revealed that the deposition process was necessary to accurately calculate the volume of soil loss and topography changes. Therefore, debris flow analysis using the numerical model and considering the deposition process would help not only to analyze the impact area, but also to accurately estimate the flow time and topography change.

As shown in Tables 2 and 3, the debris flow simulations for the Raemian and Sindonga apartment basins were successfully carried out. In addition, regardless of whether deposi-

tion was considered, the inundated depth and maximum velocity near both apartments were similarly reproduced. However, it was difficult to implement the flow velocity of the debris flow, and in particular, the sediment flowed at 19 m/s in the Raemian apartment basin, which was very low compared to the observed value of 28 m/s. Its rapid velocity was due to the fact that the debris flow event at Mt. Umyeon in 2011 contained considerably more water than the others [40,44,46]. However, the flow velocity in the Sindonga apartment basin was well-implemented. Therefore, it revealed that the Voellmy rheological equation struggles to demonstrate a flow velocity of $\geq$20 m/s [40].

In addition, the debris flow simulation for the Seongdeok basin was successfully carried out. The soil loss and impact area were well-implemented through the deposition process. Figure 10 also demonstrates the importance of the deposition process in the analysis of the debris flow. In particular, the erosion–entrainment–deposition model had a considerable influence on the modeling of the deposition that occurred in the upper middle region of the basin as well as deposition in the lower region of the basin, which is a residential area. However, the damage across the river, which did not appear in the actual event, was revealed by the simulation (Figures 10 and 11). This result was assessed to be an error caused by the fact that the DEM data are low-resolution to implement the curvature of terrain and was also caused by a limitation in the numerical model in which the sediments flowing into the river could not be implemented. However, using the simulation results in Table 4, it was estimated that the approximate damage scale and damage range were well-implemented.

In this paper, the deposition process was implemented to simulate the debris flow with the erosion and entrainment processes. In addition, deposition plays an important role in the analysis of debris flows as well as erosion and entrainment. However, the current body of research regarding deposition is relatively incomplete. Therefore, studies that explore deposition need to make better predictions of debris flows. In addition, the analysis method in this paper is not practical for unmeasured catchments because data regarding the collapse point is essential for the analysis of debris flows. Therefore, in future research, we plan to comprehensively implement slope stability and debris flow analysis to demonstrate and predict debris flow events that occur in unmeasured areas and areas with a high risk of landslides and debris flows.

## 6. Conclusions

In this study, we attempted to analyze the importance of depositional processes on the flow and accumulation process of debris flows. Among the debris flow events that occurred in Mt. Umyeon, Seoul in 2011, the Raemian and Sindonga apartment basins, and the Gokseong-gun, South Jeolla Prefecture in 2020, the Seongdeok basin was selected as the study area and the debris flows were well-implemented. To analyze the simulation results such as the flow velocity, amount of soil loss, and inundated depth, the ROC method was used. Implementing the debris flows while considering the erosion–entrainment–deposition process was successful and the importance of the deposition process was revealed through the simulation results.

In particular, deposition played a significant role in estimating the range of the damaged area caused by debris flows. In addition, the debris flows over time revealed that the deposition process helped assess the flow time. Debris flow analyses that consider erosion, entrainment, and deposition should help analyze the topography change more precisely. The overall analysis results demonstrated an approximately 0.13–0.17 (13–17%) higher accuracy when considering deposition. According to this result, analyses of debris flow events that account for the erosion–entrainment–deposition process could calculate the impact area, flow time, and topography changes. In other words, the deposition process helps simulate the debris flow so that the results are closer to the natural phenomenon along with the erosion–entrainment process.

In the Seongdeok basin, it was not possible to accurately implement the debris flow event due to the lack of on-site investigation. Therefore, it was analyzed and verified

based on the damaged area captured by drones after the debris flow. In the simulation results, we attempted to precisely analyze the effect of reducing the impact area due to the rice fields and river in the field survey, but this effort was limited to implementing the rice fields and river due to the limitations of the numerical model. However, the debris flow that occurred in the Seongdeok basin was successfully simulated. In addition, in the analysis that considered the deposition process, the impact area was simulated with approximately 0.05 (5%) higher accuracy. However, as observed in the two Mt. Umyeon basins, the deposition did not affect the flow height or flow velocity.

**Author Contributions:** Conceptualization, S.L. and H.A.; Methodology, S.L. and H.A.; Software, H.A. and M.K.; Validation, S.L., H.L. and Y.K.; Formal analysis, S.L. and H.A.; Investigation, S.L., H.A. and M.K.; Data curation, M.K.; Writing—original draft preparation, S.L.; Writing—review and editing, H.A. and M.K. All authors have read and agreed to the published version of the manuscript.

**Funding:** This research was supported by the Korea Ministry of Environment as "The SS projects; 2019002830001" and the Basic Research Project of the Korea Institute of Geoscience and Mineral Resources (Project code: 22-3412-1).

**Data Availability Statement:** Not applicable.

**Conflicts of Interest:** The authors declare no conflict of interest.

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
