# Peer review of "A Simple Deposition Model for Debris Flow Simulation Considering the Erosion–Entrainment–Deposition Process"

_remotesensing, doi:10.3390/rs14081904_

Round 1

Reviewer 1 Report

The paper presents a methodology for performing a debris-flow simulation by a 2D numerical model that uses a hyperbolic conservation form of the mass. In particular, the paper emphasizes the importance of considering the deposition effects of debris flow during numerical modeling. Although the overall analysis results are encouraging (even if in one case the verification is made on a low-resolution topography (5x5m)), I cannot appreciate the originality of the work; literature is plenty of papers that highlight the need to correctly simulate the processes of erosion and deposition is highlighted. For these reasons, I appreciate the efforts made by the authors and the good form of the paper in terms of presentation but I do not detect the novelty of the work.

Author Response

Thanks for reviewer’s comment. You are right. The deposition process has already proven its importance through many studies. But, we simulated the erosion, entrainment, and deposition processes through a simple model in this study. It is expected that the model developed in this study can be used effectively in unmeasured catchments where field investigations are challenging.

Reviewer 2 Report

This paper is interesting and takes important problem of monitoring and modeling of debris flow phenomena. Paper structure and source materials are appropriate. The proposed methodology can be implementation to similar investigation in the hot spot areas with debris flow risk. The obtained results are encouraging and allows for need next steps and confirmation their useful in practice. Only the Abstract needs correction, please to reduce this text to: aims, methods and obtained results. Some detailed comments are provided in the text (enclosed pdf).

Author Response

Thanks for your comment. We revise and supplement the manuscript following your suggestion by referring to the PDF file. Please see the attachment.

Reviewer 3 Report

The paper deals with depositional effect during debris-flow processes and aims to determine the depositional effect and the identification of debris flow risk areas. The research approach  ia based on a two dimensional model using an hyperbolic conservation form of the mass. Although the overall analysis results demonstrated an approximately 0.13-0.17 (13–17%) this increment, when considering deposition, is encouraging. I suggest to apply the method in more scenarios to standardise it.

Author Response

You are right. We suggested a simple model in this study, which is able to simulate the erosion, entrainment, and deposition process. However, as you suggest, the model requires further study on more debris flow events. Therefore, we will analyze more debris flow events using this model to verify its performance.